



**GLOBAL FATAL LANDSLIDE OCCURRENCE 2004 TO 2016**
Froude, Melanie J.[1] & Petley, David. N[1]
1.  Department of Geography, The University of Sheffield, Sheffield, S10 2TN
United Kingdom
Correspondence email: m.froude@sheffield.ac.uk
**Abstract**
Landslides are a ubiquitous hazard in terrestrial environments with slopes, incurring human fatalities
in urban settlements, along transport corridors, or at sites of rural industry. Assessment of landslide
risk requires high quality landslide databases. Recently, global landslide databases have shown the
extent to which landslides impact on society and identified areas most at risk. Previous global analysis
has focused on rainfall-triggered landslides over short ~5 year observation periods. This paper
presents spatio-temporal analysis of a global dataset of fatal non-seismic landslides, covering the
period from January 2004 to December 2016. The data show that in total 55,997 people were killed in
4,862 distinct landslide events. The spatial distribution of landslides is heterogeneous, with Asia
representing the dominant geographical area. There are high levels of inter-annual variation in the
occurrence of landslides. Although more active years coincide with recognised patterns of regional
rainfall driven by climate anomalies, climate modes (such as ENSO) cannot yet be related to
landsliding, requiring a 30+ year landslide dataset. Our analysis demonstrates landslide occurrence
triggered by human activity is increasing, in particular in relation to construction, illegal mining and
hill-cutting. This supports notions that human disturbance may be more detrimental to future landslide
incidence than climate.
**1.  Introduction**
Landslides are ubiquitous in any terrestrial environment with slopes, driven by tectonic (*e.g.* Bennett
*et al.*, 2016), climatic (*e.g.* Moreiras, 2005) and/or human (Petley *et al.*, 2007) activities.  Losses
(fatalities, physical asset damage and economic costs) occur when people and their associated
structures are exposed to landslides. The magnitude of the impact depends on the number of exposed





elements and their associated vulnerabilities; the consequences of the impacts; and the intensity of the
landslide event (Glade and Crozier, 2005).  Interest in quantifying landslide risk has developed since
the attempt by the International Association of Engineering Geology (IAEG) Commission on
Landslides to compile a list of worldwide landslide events for the UNESCO annual summary of
information on natural disasters in 1971 (UNESCO, 1973).  Although incomplete, five years of
records (1971-1975) recognised that landslides are a significant global hazard, with *c.*14% of total
casualties from natural hazards being attributed to slope failure (Varnes and IAEG Commission on
Landslides, 1984).  Since, there has been a growing interest in landslide hazard and risk assessment
(Wu *et al.*, 2015).

Key elements of the assessment of landslide risk are coherent, high quality landslide

databases and inventories (van Western *et al*., 2008; Van Den Eeckhaut and Hervás, 2012; Taylor *et*
*al.*, 2015). These provide systematically compiled lists of landslide events that have occurred over a
specific spatial scale (*e.g.* within a nation) within a set period of time, or that result from a single,
catastrophic triggering event (Hervás and Bobrowsky, 2009).  Spatio-temporal analysis of global
records of landslides have demonstrated the extent to which landslides impact on society, and have
identified geographical regions and countries most exposed (Petley, 2012).  Several different global
databases are actively maintained (e.g. the EM-DAT International Disaster Database, The NASA
Global Landslide Catalogue, and the Global Fatal Landslide Database on which this study is based),
and their merits and limitations are discussed by Van Den Eeckhaut and Hervás (2012) and
Kirschbaum *et al.* (2015).

Relative to other natural disasters, the International Disaster Database (EM-DAT) suggests

that landslides and mass movements account for 4.9 % of all natural disaster events and 1.3% of all
natural hazard fatalities between 1990 and 2015; 54% of these landslide events occurred in Asia
(Guha-Sapir *et al.,* 2018).  However, the dedicated global landslide databases indicate that global
multi-peril databases underestimate the impact of landslides on society. Petley (2012) showed that the
EM-DAT database underestimated the number of fatal landslide events by ~2000% and fatalities by
430% between 2004 and 2010, whilst Kirschbaum *et al.* (2015) showed that the EM-DAT database



underestimated the number of fatal landslide events by ~1400% and fatalities by 331% between 2007
and 2013.  For the most-part this under-reporting is associated with the perception of landslides as a
secondary hazard, with the cause of death often being recorded in connection with the primary hazard
(e.g. an earthquake rather than a co-seismic landslide), rather than the actual cause of the loss.
Past studies on global landslide distribution have focused on rainfall-triggered events,
recognising the importance of rainfall and climate in inhabited regions with steep slopes (Kirschbaum
*et al.*, 2012; Kirschbaum *et al.*, 2015). This paper not only provides a key update on the impact of
landslides worldwide, extending Petley (2012) to include landslides from 2004 to 2016, the study
considers trends in complex landslides triggered by human activity. Thereby, adding to the discussion
on climate versus human disturbance as current and future drivers of landslide incidence (Crozier,

2010).

**2.  The Global Fatal Landslide Database**
The Global Fatal Landslide Database (formerly termed the Durham Fatal Landslide Database) has
been compiled using systematic, English language based, metadata search tools that identify relevant
reports of landslide activity (including all mass movements falling within the definition of Hungr *et*
*al.* (2014) on a daily basis (Petley *et al.*, 2005; Petley, 2010; 2012). In common with other hazard
databases (Tschoegl *et al.*, 2006; Taylor *et al.*, 2015), mass media reports provide a first alert for fatal
landslide occurrence and impact. Reports are corroborated and data updated by source triangulation
using government and aid agency reports, academic papers and personal communications, as new
information becomes available. The dataset has been consistently collected and managed since 2004,
following a period of methodological development between 1 September 2002 and 31 December 2003
(Petley, 2012).  The approach is differentiated from that of Kirschbaum *et al.*, (2010; 2012; 2015)
because: (1) only landslides that cause loss of life are included; and (2) all landslides are included, as
opposed to only those triggered by rainfall. In addition, the global fatal landslide database has been
compiled over a longer period.  Although media reporting tends to be biased towards events with
human casualties (Carrara *et al*., 2003), which is favourable for a database of this nature, it is
recognised that the data collected is to some degree an underestimate of the number of fatal





landslides, and their associated losses. Events that occur in remote mountain regions, or that result in a
small number of fatalities, are less likely to be reported than multi-fatality events and/or those that
occur in urban centres (Petley, 2009). Reliability of reporting is also spatially variable, based on the
robustness of regional communication networks, which are considered more consistent in developed
nations (Petley, 2010; Kirschbaum *et al.*, 2010), and in some cases political considerations (*e.g.* very
few landslides are recorded in North Korea). The true number of fatalities may be slightly
underestimated when victims die of landslide derived injuries weeks to months following the event
(Petley, 2012). Furthermore, solely non-English reporting of events will account for some missed
reporting. However, Sepúlveda and Petley (2015) compared the Global Fatal Landslide Database with
an independently compiled database based on original Spanish and Portuguese language reports for
Latin America, and found a difference of only 5% of total records, generally associated with
landslides with small numbers of fatalities. Combined these effects may underestimate the true level
of loss by up to 15% (Petley, 2012); however the methodology of collation of the Global Fatal
Landslide Database is considered robust.
Since 2004, the database has been compiled to include the date of occurrence, the description
of landslide location; an approximate latitude and longitude for that location; the country and
geographical region (based on UN classifications, UNSD, 2018) in which the landslide occurred; the
number of fatalities and injuries; and whether the event was triggered by precipitation, seismicity or
another cause. Seismically triggered landslides in the database are excluded from analysis herein,
because the catalogue of events is not considered complete (see Petley, 2012). These equate to 168
earthquake events and 3978 fatalities. In preparation of this paper, all landslide reports were reviewed
to enhance the classification of the trigger event according to Table 1, using keyword searches in the
original text describing the landslide. The median spatial precision of entries is 681 km$^2$, with an
interquartile range of 1 to 3477 km$^2$; but all results are located to within political country boundaries.






3.  **Global Fatal Landslide occurrence, 2004 to 2016**

The total number of fatal landslides recorded worldwide, excluding those triggered by earthquakes, over the twelve calendar years between 2004 and 2016 (inclusive) was 4862. The spatial distribution of landslides (Fig. 1a and 1c) is clearly heterogeneous, with high areas of incidence in:

- Central America between Costa Rica and the South of Mexico

- The Caribbean islands.

- South America, along the Andes mountain range from Venezuela to Bolivia and to a lesser extent Chile, with another cluster of events on the east coast of Brazil around the states of Sao Paolo and Rio de Janeiro.

- East Africa, around the borders between Tanzania, Rwanda, Burundi, Kenya, Uganda and Democratic Republic of the Congo.

- Asia, which is the site of the highest number of events (75% of landslides). Substantial numbers of landslides occur along the Himalayan Arc, in states across India and southeast China, as well as high numbers in the neighbouring countries of Laos, Bangladesh and Myanmar, and southwards on islands that form Indonesia and the Philippines.

- There are smaller clusters in Turkey and Iran, as well as in the European Alps.

Fatal landslides cluster around cities (Fig. 1c) and occur most frequently in countries with lower Gross National Income (GNI, Fig. 1c) at locations known to be susceptible to landslides, based on the analysis of physical characteristics of the environment (see Hong *et al.*, 2007; Stanley and Kirschbaum, 2017). Textual analysis of landslide reports shows many events occurred in mines or quarries (423 landslides), and 568 landslides in the dataset occurred on roads. Relative poverty is also emphasised in reporting: the term 'slum' is explicitly used to describe the impacted community 29 times, while broader terms to indicate relative poverty are used 267 times within landslide reports. These observations support previous research that fatal landslides are most prevalent in densely occupied urban centres (Alexander, 1989; Anderson, 1992; Petley, 2009), along roads (Hearn, 2011;



Lee *et al.*, 2018), and at sites rich in natural resources (Zou *et al.*, 2018). In common with other
natural hazards, the poor are disproportionately affected by landslides (Hallegatte *et al.*, 2018).

Fig. 2 shows landslide occurrence in pentads, smoothed with a 25-day (*i.e.* five pentad)

moving average. The most landslide events in a single pentad was 48, in early October 2009; of these
45 were triggered in a single day (8th October 2009) by Typhoon Parma in the Philippines. Rainfall is
the leading trigger of landslides. The majority of non-seismic fatal landslides (2004-2016) in the
database were triggered by precipitation (79%). Fig. 3a shows landslides triggered by rainfall in
pentads, compared with the complete non-seismic landslide dataset. The data series are strongly
correlated (*R*, 0.933, *p-value*, 0), indicating that precipitation landslides explain 93% of the variance
of the complete dataset. Fig. 3b shows landslides that were not triggered by rainfall, and where the
trigger is known (*e.g.* mining). We term these events "complex landslides" herein. These landslides
constitute 16% of the complete dataset and present a different pattern through time when compared
with rainfall-triggered landslides. There is a notable increase in the number of landslides with
complex triggers from about 2006, which we ascribe to improved event capture.

The rainfall-triggered landslide data in Fig. 3a (and the complete landslide series, Fig. 2)

contain a strong seasonal pattern of landslide occurrence through the annual cycle, as noted by Petley
(2012). Autocorrelation measures the linear relationship between lagged values of a time series. The
autocorrelation of the rainfall-triggered pentad landslides series (Fig. S1) shows the correlation
coefficient between the original series and a lagged version of the series, where the series lags
between 1 to 948 pentads (5 days to ~13 years). The autocorrelation oscillates around 73.5 lags
(pentads), equating to one calendar year. This pattern is indicative of annual seasonality in the data.
Conversely, the autocorrelation of the complex landslides pentad series (Fig. S2) does not contain this
pattern and the correlation coefficients are generally weak. This indicates that there is no seasonal
pattern in the complex landslide series, which is to be expected in events that are not triggered by
meteorological processes.




### 3.1. Seasonality

Landslide occurrence peaks in the northern hemisphere summer, and there is notable inter-annual variation, both in the size and shape of the annual cycle. Seasonality in the global series (Fig. 2 and 3a) is associated with the annual cycle of rainfall-triggered landslides in South, South East and East Asia, and South and Central America (Fig. 4). Combined, these geographical regions contain 88% of all rainfall-triggered landslides and account for 96% of variance in the global seasonal cycle (Table B1). There is a correlation between the mean monthly rainfall and landslide series, for four of five regions (Fig. 5 and Table 2), reflecting the triggering effect of seasonal precipitation. However, the strength of relationship between seasonal patterns of rainfall and the seasonal pattern of landslides is variable between regions.  The pattern is strongest in East Asia and South Asia. This corroborates Petley (2012) who identified the strong relationship between landslide occurrence and seasonal rainfall from a shorter period of data (2004 to 2009).

Seasonal rainfall in East and South Asia is associated with the onset and withdrawal of the Asian monsoon (*e.g.* Webster, *et al.* 1998), delivered by the seasonal reversing of winds to flow from ocean to land in the summer months, resulting in the majority of annual rainfall occurring between June and September (Turner and Annamalai, 2012). In South Asia, landslide incidence increases in Nepal, India, Bangladesh, Bhutan and northern Pakistan during the summer monsoon. India and Nepal contribute 16% and 10% respectively of all rainfall-triggered landslides in the global dataset; of these 77% and 93% occurred during the summer monsoon, meaning 21% of all rainfall-triggered landslides globally were triggered by seasonal monsoon rainfall in India and Nepal. In East Asia, tropical cyclones extend the length of the rainfall season: 109 landslides were triggered by typhoons between April and October in China, Japan and South Korea, representing 16% of rainfall-triggered landslides in East Asia, and 3% of global rainfall-triggered landslides. The East Asia landslide record is dominated by events in China (81%, 503 landslides), of which 409 landslides were triggered during the summer monsoon rainfall season. China alone contributes 15% of all global rainfall-triggered landslides, although the pattern is heterogeneous.





Although, the seasonal landslide series for Central and South America do not explain much
variance in the global seasonal landslide cycle (because of the comparatively low number of
landslides), there is strong correlation between patterns of landslides in the region and patterns of
rainfall (Table 2). Central America and parts of the Caribbean experience a summer rainy season
between May and October, associated with the position of the Inter-Tropical Convergence Zone
(ITCZ; Garcia *et al.*, 2009). The season is bimodal, with peaks in rainfall either side of a midsummer
drought, between late June to August (Magaña *et al.*, 1999). The season is enhanced by the Atlantic
basin hurricane season from 1 June to 30 November (NOAA, 2018a).  The pattern of landslides
reflects these precipitation drivers. South America spans ~70º of latitude leading to local variability in
climate (Sepúlveda and Petley, 2015). The peak annual rainfall for the continent as a whole occurs
during the period from December through to February, delivered by the South American Monsoon
System (SAMS), which is driven by the position of the ITCZ to the south of the equator (Garcia *et al*.,
2009). However, in parts of south-eastern Brazil, where there is a prevalence for fatal landslides (Fig.
1), the rainy season extends into March (Rao and Hada, 1990). In northern Peru, rainfall peaks
between April and June in the west, and is bimodal in the east, with peaks in April and December
(Espinoza *et al.*, 2009). Colombia's meteorology is particularly complex due to the convergence of
the Equatorial Mid-tropospheric Easterly Jet (EMEJ) and the Choco Jet; the resulting rainfall
distribution is bimodal, with peaks in April-June and August-September, depending on precise
location and the choice of rainfall data and model (Sierra *et al.*, 2015). Most rainfall-triggered fatal
landslides in South America occur in Brazil (37%) and Colombia (32%), most notably in south-
eastern Brazil and central Colombia, and this is evident in the distribution of annual rainfall and
landslide occurrence (Fig. 5d).
The weak relationship between rainfall and landslides in South East Asia reflects the complex
weather systems operating in the region. Most landslides occurred in the Philippines (46%) and
Indonesia (32%). Typhoons caused 22% of rainfall-triggered landslides in the region, and 5%
globally; most typhoon-triggered landslides occurred in July through to October (75%), in line with
the main tropical cyclone season. In the Philippines, 42% of rainfall-triggered landslides were caused





by Typhoons, whilst the equivalent value for Vietnam was 22%, although of a much lower total. The
pattern of monsoon rainfall in Indonesia and the Philippines varies by geographical location. In the
west of the Philippines, summer monsoon occurs between June and October, while in the east, the
winter monsoon occurs between October and March (Kubota *et al.*, 2017). This pattern is evident in
the distribution of rainfall-triggered landslides in the Philippines (Fig. 1a). The onset and termination
of the monsoon in Indonesia varies from September to June in north Sumatra and late November to
late May in east Java (Naylor *et al.*, 2007). Consequently, 72% of rainfall-triggered landslides occur
between November and April, when the majority of Indonesia is experiencing monsoon rainfall. The
peak in landslide activity relative to rainfall in August to October in South East Asia (Fig. 5b) is
mainly due to the localised typhoon rainfall not captured in the regional rainfall average.

### 3.2. Medium term trend in landslide occurrence

There was a general increase in recorded landslide occurrence between 2004 to March 2010, followed
by a general decrease in landslide occurrence through to April 2015, after which landslide incidence
has generally increased (Fig. 6a). Petley (2012) identified improvements in the reporting of single
fatality landslides as contributing to the general increase in events in the fatal landslide record from
2004 to 2010. The number of fatalities resulting from non-seismic landslides between 2004 and 2016
was 55997. Fig. 6b, shows that the pentad series of fatality is very noisy; the data do not contain an
increasing or decreasing trend, nor are there distinguishable medium-term peaks in the data. Very few
landslides generated more than 1000 fatalities (0.1%), and only one landslide resulted in more than
5000 fatalities. This was the Kedarnath landslide in June 2013 in Uttarakhand state, India, which was
caused by extreme meteorological conditions that generated flooding and two large landslides in a
mountainous area occupied by thousands of religious pilgrims (Allen *et al.*, 2015).
Landslides by the number of fatalities are grouped by the infinite series (1, 2, 4, 8, 16…).
There is a significant increasing trend in single-fatality landslides (Fig. 7c); 29% of landslides were
single-fatality events. There is also a weaker decreasing trend in landslides resulting in 64 to 128
fatalities (Fig. 7d); 1% of landslides were in this group. No other grouping contained a significant
trend with time. Both the single fatality and 64 to 128 fatality series are above the regression line in



2010 (Fig. 7c and 7d). Removing these two groups from the global series (Fig. 7e) it is evident that
single fatality events enhanced the peak around 2010, and in 2016.
By year, different geographical regions experience above/below average landslide activity
(multi-fatality landslides, Fig. 7f; single-fatality landslides, Fig. 7g). In 2005, 2009, 2010 and 2011,
several regions experienced greater than average landslide occurrence simultaneously. The high
impact of landslides globally in 2010, has been discussed by previous authors (Kirschbaum *et al.*,
2012; 2015; Petley, 2012; Sepúlveda and Petley, 2015). The peak in landslide activity was generated
by anomalous landslide occurrence in several regions simultaneously (Fig. 7f and 7g), but overall the
geographical pattern of rainfall triggered landslides in 2009 and 2010 reflects the occurrence of a
moderate El Niño in 2009 and a moderate La Niña in 2010 (NOAA, 2018b).
In Central America, Kirschbaum *et al.* (2012) showed that precipitation was significantly
above average in the summer months in 2010, particularly in September. This increase was linked to
the known impacts of La Niña events on tropical cyclone frequency and track (*e.g.* Elsner *et al.*, 1999;
Curtis *et al.*, 2007). By number, 2010 was the year in which the most landslides (17 events, compared
with an average 6 events per year), were directly associated with tropical cyclones in reports or related
to storm tracks (based on NOAA, 2018c). Although, these landslides only equate to 35% of all
rainfall-triggered landslides within 2010, the remaining 65% of events, not triggered by a tropical
cyclone all occurred during the hurricane season (May to November), likely due to unsettled weather
associated with the passage of large storms in the region. Central America receives tropical cyclones
from the Atlantic basin and the North Pacific basin (NOAA, 2018c). Storms from the Atlantic basin
may make landfall along the eastern coastline of Central America and travel inland, occasionally
retaining enough energy to cross over into the Pacific. Storms that have crossed over basins or new
storms, which have formed in the Northeast Pacific basin, may make landfall on the western coast of
Central America. Not only were the frequency of landfalling tropical storms and hurricanes elevated
from both basins in 2010, but the track of these storms intercepted populated areas in steep terrain
(NOAA, 2018c). The majority of rainfall-triggered landslides in Central America in 2010 were in
Mexico and Guatemala (43% and 37%, respectively). In Guatemala, eight landslides were triggered



by Tropical Storm Agatha in late May 2010, causing 182 fatalities. Four landslides were associated
with Hurricane Alex which travelled up the east coast of Guatemala, Honduras and then inland to
Mexico in late June-July 2010. Hurricane Karl then made landfall on the east coast of Mexico in
September: two landslides are associated with this storm (killing 12), but a succession of fatal
landslides in the states of Oaxaca, Chiapis and Puebla through which the hurricane passed, were noted
in the weeks following the storm.
Sepúlveda and Petley (2015) observed a weak correlation between La Niña conditions in late
2010-2011 and heightened landslide activity in Colombia and Venezuela. Considering a longer time
series (2004 to 2016), this study identifies above average landslide activity in several nations in South
America in 2009 and 2011. In Brazil, 54% of all rainfall-triggered events occurred between 2009 and
2011. Activity peaked in December 2009 to April 2010 (El Niño) and January 2011 (La Niña),
corresponding with the seasonal ENSO precipitation patterns observed by Grimm and Tedeschi
(2009). The number of landslides in Venezuela and Colombia between 2009 and 2011 peaked in
November 2010, associated with positive rainfall anomalies during the austral summer La Niña
(Tedeschi *et al.*, 2013).
The majority of landslides in East Asia occur in China (83%); in 2010, 87% of all rainfall-
triggered events were located in China, and rainfall-triggered landslide occurrence (67 landslides) was
above the mean (45 landslides). From a shorter period of observation, Kirschbaum *et al.* (2012)
identified a high incidence of rainfall-triggered landslides (fatal and non-fatal) in central eastern China
in 2010: particularly in July and August corresponding with a peak in rainfall. Rainfall-triggered
landslides were above average for most months in 2010 in China, but May through to September was
very active (57 landslides compared with an average 38). The East Asian subtropical summer
monsoon (a component of the East Asian monsoon) has a significant effect on seasonal variations in
rainfall across China (He and Liu, 2016), and precipitation patterns alter in response to ENSO
conditions (Yang and Lau, 2004; He et al., 2007; Zhou *et al.*, 2014).
In China in 2010, there were fewer than average landslides triggered by tropical cyclones
from the Northwest Pacific basin. There was low typhoon activity due to the rapid transition from the



2009/2010 El Niño to the 2010/2011 La Niña, which altered airflows in the North West Pacific basin
(Kim *et al.*, 2012). Conversely, in the Philippine domain, tropical cyclone occurrence was above
average in July to December 2009 (Corporal-Lodangco *et al.*, 2015). During the northern hemisphere
summer months of an El Niño, the genesis location of tropical cyclones shifts eastwards (Chan 1985;
Chan 2000; Chia and Ropelewski, 2002). In these conditions, cyclones travel further before they may
make landfall, enabling them to strengthen (Camargo and Sobel, 2005), and there is a tendency for
more storms to affect the northern-central Philippines (Lyon and Camargo, 2009). In 2009, 67% of
rainfall-triggered landslides in the Philippines were associated with tropical cyclones: 60 landslides
compared with an average 12 triggered by tropical cyclones. As noted previously, many of these were
triggered on the same day (8 October 2009) by Typhoon Parma.
Although the peak in landslides in South East Asia in 2009 is dominated by typhoon triggered
landslides in the Philippines, there was an increase in landslides in Indonesia (33 landslides compared
with an average of 24 per year); of these 24 events were triggered by rainfall, 8 by mining and one
trigger was not known. Rainfall-triggered landslides were very slightly above average in Indonesia in
2009 but it was the events triggered by human activity, which contributed most to the anomalous
landsliding in Indonesia. These landslides are discussed in the next section.
Between 2004 and 2016, four El Niño events occurred: weak El Niño (2004/2005,
2006/2007), strong El Niño (2009/2010) and very strong El Niño (2014/2016; NOAA, 2018b). Weak
La Niña was observed in 2005/2006, 2008/2009, 2016, and strong La Niña in 2007/2008 and
2010/2011 (NOAA, 2018b). There does not appear to be a consistent relationship between ENSO
phase and the regional distribution of landslides, although elevated regional rainfall (and thus
landslides) has been associated with ENSO sea-surface temperature (SSTs) anomalies. The peak in
landslides in Central America in 2005 is composed predominantly of tropical storm and hurricane
triggered landslides in El Salvador, Mexico, Guatemala and Honduras. The 2005 North Atlantic
Hurricane season was the most active since records began in 1851, driven by high SSTs in the
Tropical North Atlantic (10°-20°N) linked with global warming and the 2004/2005 El Niño
(Trenberth and Shea, 2005). Landslides were also above average in 2005 in East Asia: most events



occurring in China, triggered by monsoon rainfall. In South Asia, landslides peaked in 2007, 2014 and
2016, the majority associated with monsoon rainfall in Bangladesh, India, Nepal and Pakistan.
Variability in rainfall from the South Asian monsoon is related to the interaction between SSTs in the
Indian Ocean dipole (IOD) and ENSO (*e.g.* Ashok *et al.*, 2007; Lu *et al.*, 2017).

The complexity of climate systems means it is not possible to draw conclusions on the

relationship between climate mode and landslide occurrence from this 13 year global dataset.
However, longer local records show promise at unpicking the impact of climate cycles on landslides.
### 3.3. Complex Triggers
Of the 4862 non-seismic landslides in the complete database, 770 (16%) were generated by a complex
trigger, and resulted in a total 3725 fatalities (Fig. 7). The majority of landslides were triggered by
mining (232 multi-fatality landslides, 67 single-fatality landslides), construction (170 multi-fatality
landslides, 140 single-fatality landslides) or illegal hillcutting (60 multi-fatality landslides, 27 single-
fatality landslides); and the majority of fatalities in all cases were people at work (90%, 76% and 84%
respectively). Globally there is a statistically significant increase in events by these three triggers (Fig.
8a, 8b and 8c); multi-fatality landslides are differentiated from single-fatality events, which increased
with time independent of trigger (Fig. 6c). By country, most construction triggered landslides
occurred in India (28%), followed by China (9%), Pakistan (6%), the Philippines (5%), Nepal (5%)
and Malaysia (5%; Fig. 9a). On average construction triggered landslides killed three people, however
a particularly severe landslide in Shenzhen, China in December 2015 killed 77 people. The event
involved the collapse of construction waste on worker quarters in an industrial site. Interestingly, the
context in which the landslides occur differs between countries. In China, the majority of events
(52%) occur in urban construction sites, while very few landslides occur on roads (7%). Conversely,
in India and Nepal, 30% and 43% of landslides triggered by construction occurred on roads.

Transportation is a "crucial driver of development" (World Bank, 2018b); however, in

mountain regions roads are closely connected with landslide risk (Lennartz, 2013). The road network
in Nepal has quadrupled in length over the last 18 years (Govt. of Nepal, 2016), and in India it has





nearly tripled in length in 24 years (Govt. of India, 2016). Population growth is frequently
accompanied by the expansion of infrastructure and settlements (Gardner and Dekens, 2007), and this
is true in India and Nepal, which have grown by ~7% between 2010 and 2015 (World Bank, 2018a).
Both countries are on a trajectory to expand their national road networks further. Increased landslide
activity in the Himalayan region has been associated with road construction (Ives and Messerli, 1989;
Haigh *et al.*, 1989; Valdiya, 1998; Barnard *et al.*, 2001; Petley *et al.*, 2007; Sait *et al.*, 2011; Singh *et
al*., 2014). Hearn and Shakya (2017) highlighted that road construction without proper route choice,
engineering design and management of spoil, increases landslide susceptibility. Fatal landslides
triggered by road construction indicate that excavation may not always be undertaken with due care
and appropriate slope engineering. Furthermore, the coincidence of construction worker and road user
fatalities from the same landslide suggests that there is pressure to keep roads under construction
open. Ives and Messerli (1989) emphasised the economic impact when roads are closed.

Between 2004 and 2016, China experienced a 6% growth in population to 1.379 billion, and a

16% rise in the proportion of the population living in urban areas (World Bank, 2018a). Urban growth
in China is driven by political policy for economic growth; economic reforms from 1978, opened
China's markets to foreign investors and relaxed migration controls, prompting rapid rural-urban
migration (Ma, 2002; Anderson and Ge, 2004). Although urbanization is encouraged by China to
increase domestic consumption, urban growth is often uncontrolled (Fang and Pal, 2016), leading to
rapid land conversion, dispersion and fragmentation of development (Schneider and Woodcock,
2008). Critically, many of China's largest cities are bounded by mountains, and urban sprawl is
encroaching on land unsuitable for development (Yu and Li, 2011). Reports in the database indicate
that fatal landslides in urban construction sites in China often occurred when engineered cut slopes
failed above the construction site (*e.g.* Zhang *et al.*, 2012), from improper construction of foundations
leading to building collapse before completion (*e.g.* Srivastava *et al.*, 2012), or from mismanagement
of construction and demolition waste (*e.g.* Yang *et al.*, 2017). In these entirely preventable
circumstances, explicit national regulation and enforcement should reduce construction related
landslide impact in China.



The increase in events triggered by mining is driven by the increase in landslides triggered by
illegal or unregulated extraction (Fig. 8d); landslides triggered by legal mining (Fig. 8e) or where the
legitimacy of the mining is unknown (Fig. 8f), do not show a statistically significant trend. By
country, India (12%), Indonesia (11.7%), China (10%), Pakistan (7%) and Philippines (7%) contribute
most to the record of landslides triggered by mining (Fig. 9b). Fatal landslides triggered by illegal
mining practises have occurred in 32 countries (Fig. 9c). By number of events, Indonesia (24) and
India (15) rank the highest, however by number of fatalities Myanmar (403 fatalities from 9
landslides) stands out. Shifts in spending power and the infusion of the internet and smart-technology
in daily life have driven an exponential increase in the consumption of electronics, placing pressure
on the demand for rare earth elements (Dutta *et al.*, 2016). Furthermore, growth in the precious stone
market fuelled both by economic uncertainty, and a growing middle class in Asian nations such as
China, where gemstones are a key part of cultural heritage (The Economist, 2011), is thought to have
led to an increase in the number of small-scale mining operations globally (Hruschka and Echavarría,
2011), and the upscaling of small-scale mines to larger scale operations. Fatal landslides in Myanmar
(Burma) have significantly increased because of the unregulated expansion in jade mining within the
Kachin state. Critically, the high value of jade and lack of enforced operator accountability appears to
be driving poor mining practises, which place workers and local residents at risk of slope collapse
(Global Witness, 2015). Demand for rare earth elements and gemstones are thus driving an increase in
mining-related landslides, with the potential for landslide occurrence to rival that associated with rural
road expansion.
Cutting slopes for the purposes of obtaining earth surface materials, or to alter slope geometry
during construction, may result in slope failure if the site is not properly engineered. The term hill-
cutting is used here in relation to discrete slopes that have been altered without permission for the
purposes of small-scale construction, earth material extraction or agriculture. Hill-cutting is most
strongly associated with urban areas in Bangladesh in the academic literature (*e.g.* Chittagong;
Ahmed, 2015 or Syhlet; Islam *et al.*, 2006). In the fatal landslide database it is an increasing problem
in Bangladesh, India and Nepal (Fig. 8c and 9d). Most fatalities occurred as people collected hillslope





materials for construction of their housing in rural communities, and reports indicate those involved
were from poor families living in informal settlements. In total, 11 of the 87 landslides were directly
related to the practice of using hillslope coloured clay for the decorative coating of houses for a
religious festival; of these, nine occurred in Nepal. Critically, children are often caught up in slides
triggered by hill-cutting in Nepal: at least 40% of landslide victims were children, while a further 25%
of victims were a combination of adults (predominantly women) and children working together.
Conversely, in Bangladesh the majority of victims were adults (78%) of which 79% were male. In
Nepal, India and Bangladesh, clay is an important local building material for housing, particularly in
settlements not connected to the road network, but there is a legal framework in Bangladesh to
prevent hill-cutting (Building Construction Act 1952, 1990 and the Bangladesh Environmental
Conservation Act 1995; Murshed, 2013). Building codes in Nepal provide basic guidance on slope
stability, specifically slope excavation, identification of slope instability and construction of
foundations (DUDBC, 1994); however residents in rural communities may not have access to this
information and be unaware of the hazard (Oven *et al.*, 2008). Furthermore, in India it was noted that
building regulations do not account for the geo-environmental context of the settlement, sometimes
lack clarity, and are difficult to uphold due to a shortage of technical experts and inadequate provision
to stop illegal activity (Kumar and Pushplata, 2014).

While this section discusses fatal landslides triggered by human activity, many rainfall-

triggered landslides occur on slopes which have been modified during construction (82 landslides),
agriculture and forestry (45 landslides) and mining (123 landslides); or at sites where storage of waste
has not been poorly managed (16 landslides). Of course, it is expected the majority of fatal landslides
(94%) will occur within settlement boundaries or along infrastructure, however it is evident from this
database of events that human action damages slopes increasing their susceptibility to fail.
**4. Discussion and Conclusion**
With the benefit of a 13 year time series, this study builds on past analyses of the Global Fatal
Landslide dataset, providing not only an update on the spatial and temporal distributions of landslide
impact, but also serving to highlight the importance of annual climate variability in specific landslide



prone regions on the global record. In addition, it provides new insights into the impact of human
activity on landslide incidence. The data does not indicate a discernible long-term increase or decrease
in global landslide impact; rather the record shows that there is considerable inter-annual variability in
global landslide incidence. The more active years have been associated with recognised regional
patterns of rainfall, in part driven by global climate anomalies, but there is no simple relationship
with, for example,  the El Niño Southern Oscillation (ENSO).  Relating climate modes to patterns of
landsliding is challenging because of climate complexity and change, requiring 30 year + datasets.
Future work bridging advances in climate science on regional impacts from ENSO diversity, with
local patterns of landsliding, in acutely affected areas such as India, China and Nepal, will provide
useful models for forecasting seasonal rainfall distribution and landslide impact.
Our analyses have demonstrated that landslide occurrence triggered by human activity is
increasing, in particular in relation to construction, illegal mining and illegal hillcutting. Human
disturbance (land use change) may be more detrimental to future landslide incidence than climate
change (Crozier, 2010; Anderson and Holcombe, 2013), and this is evidenced by a number of studies
(Innes, 1983; Glade, 2003; Soldati *et al.*, 2004; Imaizumi *et al.*, 2008; Borgatti and Soldati, 2010;
Lonigro *et al.*, 2015). Fatal landslides occur when construction and mining: (1) do not apply
appropriate slope engineering, (2) mismanage spoil and, (3) do not undertake a feasibility assessment
(Hearn and Shakya, 2017). Appropriate building regulations that account for the geo-environmental
context of the settlement, provide clear guidance on engineering and are enforced by local technical
experts, are paramount in managing landslide risk associated with urbanization and natural resource
exploitation.
Holcombe *et al.* (2016) emphasised that planning policy alone is not sufficient to control
landslide risk in developing nations, because of the rapid and informal nature of construction, and
low-income of residents, who cannot finance expert guidance when building their homes. Settlements
are often built on hazardous land around urban centres and on roadsides, because of the benefits of
service access and employment opportunities (Smyth and Royale, 2000; Oven *et al.*, 2008; Lennartz,
2013; Anhorn *et al.*, 2015). Hillcutting is the dominant driver of instability during informal
construction (Holcombe *et al.*, 2016), and our results indicate fatal landslides triggered by hillcutting





are increasing in Bangladesh, India and Nepal. Several landslides were triggered when people cut
slopes to collect coloured clay to decorate their houses for religious festivals. Here, simple
communication of landslide risk by local non-governmental organisations (NGOs) could prevent
future fatalities from this practice. Where governments are limited in capacity at a local level, NGOs
are important in implementing disaster risk reduction (Jones *et al.*, 2016), such as supporting
community-based slope engineering (*e.g.* Mossaic; Anderson and Holcombe, 2006).

Reporting of fatal landslides is likely to increase with the global growth in mobile technology

and internet access. Furthermore, advances in web mining (data retrieval from the internet based on
search criteria) and text mining (transforms unstructured data into structured to discover knowledge)
using machine learning offer methods to improve capture of landslide reporting and data evaluation
(*e.g.* Bhatia and Khalid, 2008; Kumar et al., 2018). Continued collection of the database will develop
our understanding of the effect of climate and human disturbance on global landslide impact. The
dataset is a useful tool in identifying acutely landslide prone parts of the world and specific local
drivers of landslide impact; thereby highlighting locations which would benefit from further
development in early warning technology, landslide risk assessment and community capacity
building; in support of the future directions of the International Consortium on Landslides (Alcantara-
Ayala *et al.*, 2017).





**Figures**
**Figure 1.** (**a**) The location of non-seismically triggered fatal landslides from AD 2004 to 2016.
Individual landslides shown by a black dot. (**b**) Number of non-seismically triggered fatal landslides
from AD 2004 to 2016 by country. (**c**) The Gross National Income per capita (US $) by country
(World Bank, 2018), and the location of major urban centres globally (ESRI, 2018).
**Figure 2.** The occurrence of non-seismically triggered landslides from AD 2004 to 2016, and
cumulative total of recorded events. The data are arranged by pentads (five-day bins), starting on the
1st January each year, thus the first pentad includes records for 1-5 January, and there are a total 73
pentads. A simple 25-day moving average is shown.
**Figure 3.** (**a**) The occurrence of rainfall triggered landslides from AD 2004 to 2016 (blue). The data
are arranged by pentads (five-day bins), starting on the 1st January each year. A simple 25-day
moving average is shown. The 25-day moving average for all non-seismically triggered landslides, is
shown in black. (**b**) The occurrence of complex landslides from AD 2004 to 2016 (purple). The data
are arranged by pentads (five-day bins), starting on the 1st January each year. A simple 25-day
moving average is shown. The 25-day moving average for all non-seismically triggered landslides, is
shown in black.
**Figure 4.** Mean number of landslides per pentad through the annual cycle for all rainfall-triggered
landslides, and by geographical region. The 20th pentad is the 6-10 April, the 40th pentad is 15-19
July and 60th pentad is 23- 27 October.
**Figure 5.** Mean daily rainfall (mm) by month between AD 2004 and 2016 summarised by
geographical sub-region (blue bars). Global Precipitation Climatology Centre data (Xie *et al.*, 2013;
GPCC, 2018) was processed in ESRI ArcMap and Matlab. Mean daily rainfall-triggered landslide
occurrence by month between AD 2004 and 2016 (black line). Daily values are used to overcome the
difference in month length.
**Figure 6.** (**a**) The occurrence of non-seismically triggered landslides from AD 2004 to 2016: 25 day
and 1 year moving average (see also Fig. 2). (**b**) The number of fatalities from non-seismically



triggered landslides from AD 2004 to 2016 by pentad with 25 day moving average. (**c**) Number of
single fatality landslides AD 2004 to 2016. (**d**) Number of landslides incurring 64 to 128 fatalities per
event from AD 2004 to 2016. (**e**) Comparison of the complete landslide series (Fig. 6a) and multi-
fatality landslide series (excluding the 64 to 128 fatality class). (**f**) Anomalies in landslide occurrence
by year by geographical region (multi-fatality events). (**g**) Anomalies in landslide occurrence by year,
by geographical region (single fatality events). Values greater than 1 standard deviation from the
mean are shown by a grey circle. Values greater than 2 standard deviations from the mean are shown
by a black circle.
**Figure 7.** Distribution of triggers of complex landslides (770 events).
**Figure 8.** Number of landslides triggered per year by (**a**) construction, (**b**) mining, (**c**) illegal hill-
cutting, (**d**) illegal mining, (**e**) legal mining and (**f**) mining (not specified). The black series contains
only multi-fatality landslides. The grey series contains single and multi-fatality landslides.
**Figure 9.** By country, the number of landslides triggered by (**a**) construction, (**b**) mining, (**c**) illegal
mining and (**d**) hill-cutting, between A.D. 2004 to 2016.




**Data availability**
The global fatal landslide database (2004 to 2016) is not currently available to the public, however a
web-platform is under-development to host the data openly at the University of Sheffield, UK.

**Appendix A**
**Figure A1.** Sample autocorrelation plot for the pentad rainfall-triggered landslides. The 99%
confidence interval is shown by the blue horizontal lines.
**Figure A2.** Sample autocorrelation plot for the pentad complex landslides. The 99% confidence
interval is shown by the blue horizontal lines.
**Appendix B**
**Table B1.** Hierarchal linear regression results comparing the impact of seasonality in geographical
regions with the global mean number of landslides per pentad through the annual cycle (see Fig. 4).
The data series for each geographical region are sequentially added into the regression (such that the
second row of the table is a regression of S.Asia + S.E. Asia with the global series).

**Author contribution**
DP developed the methodology (2002-2003) and has consistently collected the database since 2004.
MF analysed the data and wrote up the results for this submission. DP contributed to writing.
**Competing interests**
The authors declare that they have no conflict of interest.
**Acknowledgements**
The research has been supported by the NERC/Newton Fund grant NE/N000315/1 and
NE/N000315/2.



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

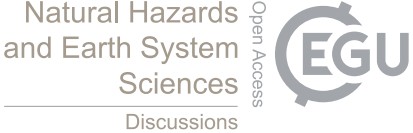



**Tables**
**Table 1** Landslide trigger classification.

| Classification | Definition | Keyword search terms |
|---|---|---|
| unknown | No trigger or obvious cause specified | - |
| precipitation | Rainfall raises pore-pressure in slope materials triggering failure. | "rain", "sleet", "storm", "hurricane", "precipitation", "flood", "water", "torrent" |
| earthquake | Strong ground motion associated with earthquakes weaken slope materials triggering failure (coseismic landslides). | "earthquake", "aftershock", "seismic", "tremor" |
| illegal mining | Unregulated or informal mining of slope materials in designated quarry or mine, where permission to extract material has not been granted. | "illegal", "permit", "regulat", "close", "informal", "pick", "illicit", "abandoned", "traditional", "license", "ban", "mine", "quarry", "spoil", "pit", "excavat" |
| illegal hillcutting | Hillcutting refers to the process of removing material from a hillslope for the purposes of altering its shape and/or to obtain slope material for use in construction, manufacture or farming. It is differentiated from mining, because it occurs on slopes that are not within a designated site of mining or quarrying, instead hillcutting typically occurs on individual slopes on steep agricultural land or on man-made slopes such as those along transport routes. Hillcutting differs from construction, because slope modification does not follow an engineering design to ensure slope stability. Hillcutting is assumed to be undertaken in an informal, unregulated manner (this is frequently noted in landslide reports) | "hillcut", "illegal", "permit", "regulat", "informal", "illicit", "traditional", "license", "ban", "excavat" |
| legal mining | Regulated and/or permitted mining of slope materials in designated quarry or mine, where permission to extract material has been granted and operations are managed. | "legal", "permit", "regulat", "pick", "license", "mine", "quarry", "spoil", "pit", "excavat" |
| mining (unknown) | Slope materials are extracted from a designated quarry or mine, but the report does not make it clear if the extraction is permitted or not. | "quarry", "mine", "spoil", "pit", "excavat" |
| construction | Permitted modification of a slope for the purposes of a construction project undertaken by professional labourers, following planning approval | "excavat", "construction", "site", "road", "build", "dig", "labour" |
| conflict and explosion | Landslide triggered by the detonation of an explosive device during military combat | "bomb", "mine", "soldier", "army", "explode", "explosion", "war", "conflict" |
| leaking pipe | Utility pipes carrying water that have been damaged and leak water onto a slope surface or within the hillslope, compromising its stability. | "pipe", "leak", "burst" |
| garbage collapse | Collapse of piles of municipal waste onto people, where stability of waste piles was disturbed by the passage of a person or persons | "waste", "trash", "rubbish", "garbage", "dump", "pick" |
| recreation | Triggered by passage of a person or persons walking/ climbing over a hillslope for recreation. | "climb", "mountain", "expedition", "ascent", "trek" |
| human action (unspecified) | Landslide report refers to a person or people present on a hillslope that collapses, without specifying the reason people occupied the slope or the landslide trigger | "people", "person", "men", "women", "children", "occup" |
| animal activity | Occupation of slope by animal triggering failure, either by weight and movement of animal on slope surface, or by burrowing within the slope subsurface. | "animal", "burrow", "tunnel" |
| fire | Naturally occurring or man-made fires, typically occurring in dry climates on vegetated terrain. | "fire" |
| natural dam or riverbank collapse | Collapse of a riverbank or natural dam without an apparent trigger, but likely caused by pore pressures building over time to a critical threshold in response to water levels. Material typically fails into a body of water, and often generates a flood wave. | "river", "bank", "dam", "earth", "flood", "wave", "collapse" |
| freezing | Heavy snowfall and expansion of water in hillslopes due to freezing, acting solely or together to destabilise the slope. | "snow", "extreme", "freeze", "ice", "cold" |
| freeze thaw (temperature change cold to hot), snowmelt | Failure of slope materials in response to temperature rise, including landslides triggered by the melting of snow or permafrost (in a non-volcanic setting) | "snow", "melt", "permafrost", "spring", "temperature" |
| volcanic eruption | Landslides (and mudflows) occurring in a volcanic environment triggered by volcanic activity- such as explosions and volcano-tectonic seismicity. This does not include events in active volcanic environments triggered by rainfall. | "volcan", "seismic", "activity", "eruption" |
| marine erosion | Triggered by sea erosion (only)- repeat wave impact | "coast", "sea", "erode" |








**Table 2.** Spearman's rank correlation between mean daily rainfall and mean daily landslides by
month (see Fig. 5).

| Region | Correlation Coefficient | P value |
|---|---|---|
| Central America | 0.8153 | 0.0012 |
| South America | 0.8062 | 0.0015 |
| South East Asia | 0.17 | 0.5974 |
| South Asia | 0.996 | 0 |
| East Asia | 0.9701 | 0 |


**Table B1.** Hierarchal linear regression results comparing the impact of seasonality in geographical
regions with the global mean number of landslides per pentad through the annual cycle (see Fig. 4).
The data series for each geographical region are sequentially added into the regression (such that the
second row of the table is a regression of S.Asia + S.E. Asia with the global series).

| Predictor Variables | N (cumulative) | % (of total N) | $R^2$ | $\Delta R^2$ |
|---|---|---|---|---|
| + S. Asia | 1295 | 31.50 | 0.4962 | |
| + SE. Asia | 2121 | 52.27 | 0.7365 | 0.2403 |
| + E. Asia | 2804 | 71.88 | 0.8618 | 0.1253 |
| + S. America | 3145 | 82.25 | 0.9129 | 0.0511 |
| + C. America | 3340 | 88.03 | 0.9575 | 0.0446 |






**Fig. 1**





**Fig. 2**

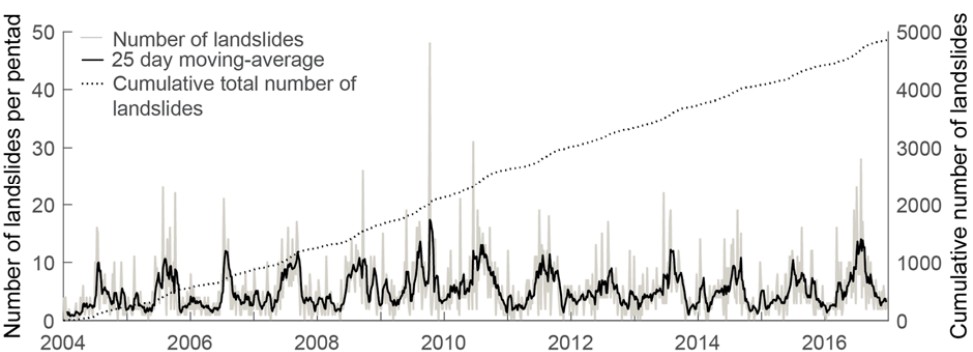



**Fig. 3**

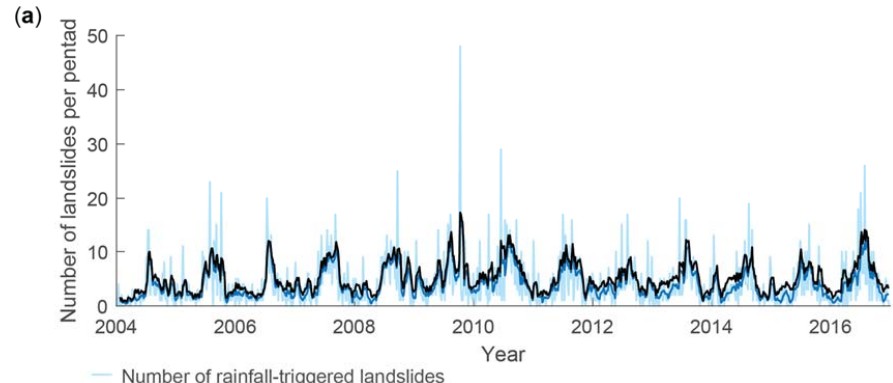

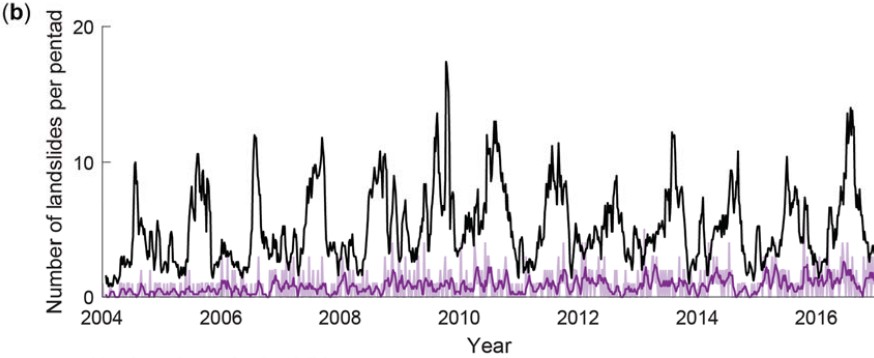


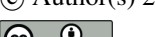


**Fig. 4**

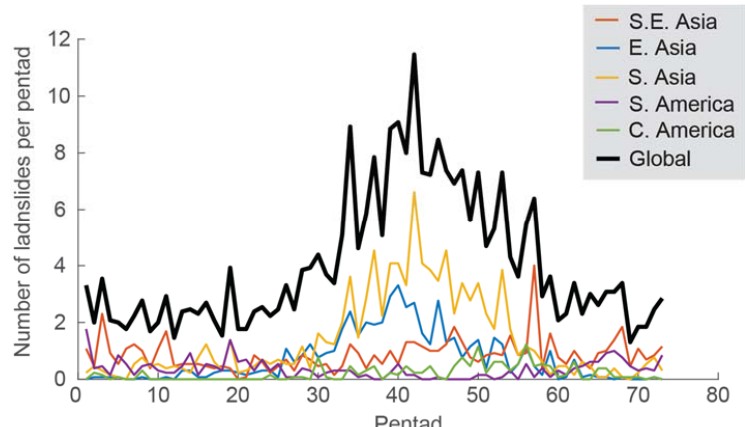


**Fig. 5**

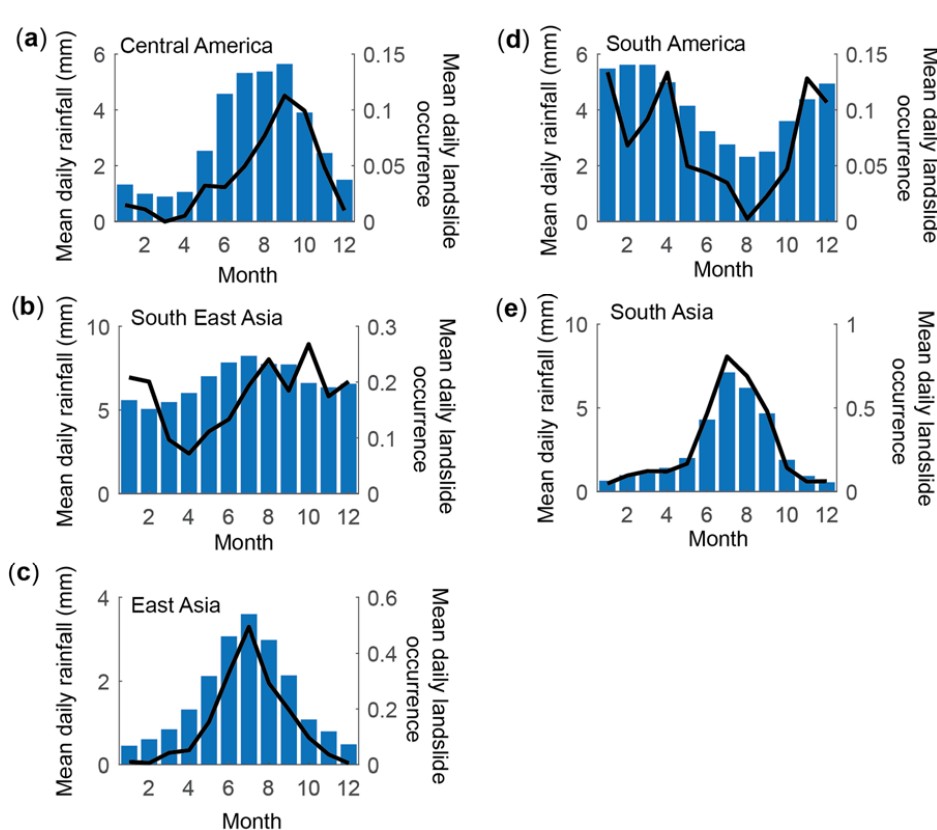





**Fig. 6**



**Fig. 7**

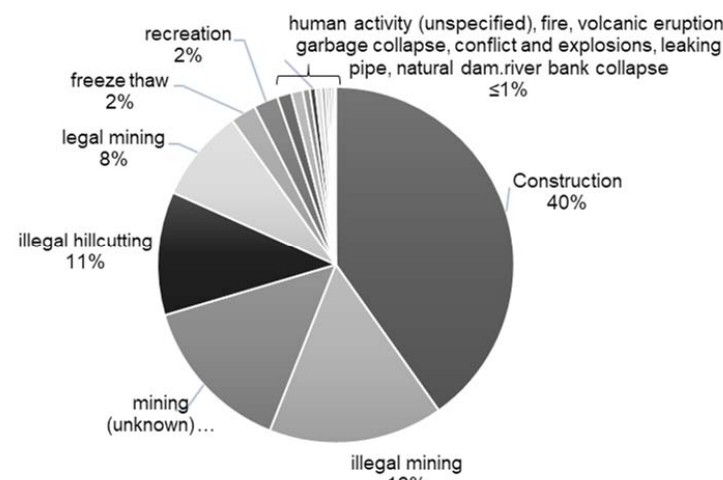
















**Fig. 8**

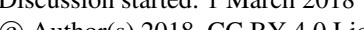







**Fig. 9**








**Appendix Figures**
**Fig. A1**

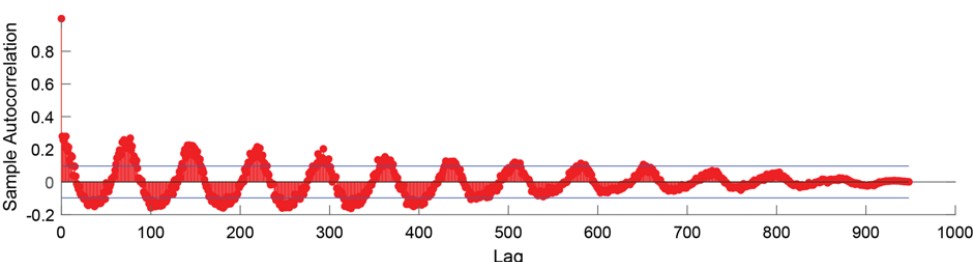


**Fig. A2**

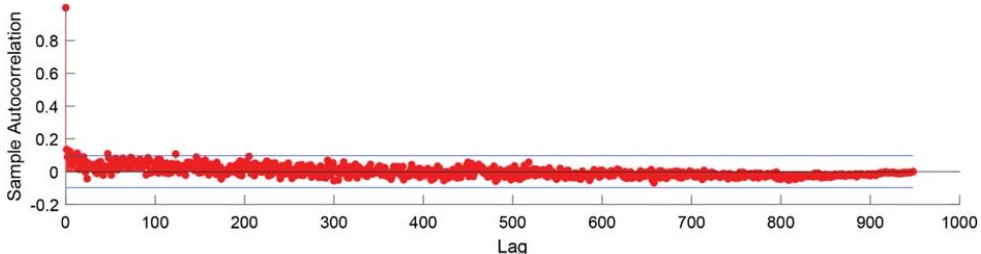
