# Peer review of "GLOBAL FATAL LANDSLIDE OCCURRENCE 2004 TO 2016 2 Froude, Melanie J.1 & Petley, David. N1 3 4 1. Department of Geography, The University of Sheffield, Sheffield, S10 2TN 5 United Kingdom 6 Correspondence email: m.froude@sheffield.ac.uk 7 8 Abstract 9 Landslides are a u"

_Natural Hazards and Earth System Sciences, 2018_

## Short Comment (SC1) · 4 Mar 2018

[revised manuscript text omitted]

**Appendix Figures**

**Fig. A1**

[Figure]

**Fig. A2**

[Figure]

---

## Referee Comment (RC1) · Anonymous Referee #1 · 23 Mar 2018

Summary: Overall this manuscript is well written, provides interesting findings reviewing over a decade of landslide reports and adds value to this topic within the landslide community. With minor revisions, I recommend this manuscript for publication. I do not see any major weaknesses in the methodology or evaluation methods and appreciate the insight provided in the discussion section regarding the challenge with anthropogenic modification of hillslopes in causing landslide fatalities; however the authors state in several places that human disturbance may be more detrimental to future landslide incidences than climate. I think a more accurate statement is that it may be more detrimental to future fatal landslides incidences, as the understanding of how climate may modulate change in extreme rainfall and how that may impact landsliding has not, to my knowledge, been robustly addressed. Please see specific comments below.

[Figure]

Line 106-7: How was the spatial precision determined? While this does not affect the analysis in the paper, it would be helpful to provide another sentence or a reference to how this precision was obtained for each report. Line 171: What precipitation data did you use in your analysis? Line 195: I believe there is a word missing from the sentence that starts with "The season is bimodal..." possibly "with peaks in rainfall on either side of a midsummer..." Lines 240, 244, 247, 251: Should be changed to Figure 6, not Figure 7 Line 261: What is meant by the term unsettled weather? Line 312: Suggest changing "which" to "that" and removing the comma Lines 439-441: Run on sentence, consider revising. Line 444: Consider adding "fatal" to the wording, "disturbance (land use change) may be more detrimental to future fatal landslides incidence..." Line 453-55: Suggest reworking sentence, it reads rather awkwardly Line 460-461: Sentence starting with "Several landslides were triggered..." is a repetitive sentence and is likely not needed here. Line 461: Suggest omitting the word "simple" as often effective communication of risk by the government is not that simple. Lines 471-476: Suggest reworking sentence, it reads rather awkwardly Line 479 and other references: is it necessary to write AD prior to the years 2004 to 2016? I suggest removing as I am confident readers will not confuse it with events 4000 years ago.

---

## Short Comment (SC2) · 9 Apr 2018

Dear, First of all, congratulations for your iniciative. In 2011 in Rio de Janeiro State, over than 950 people die due a massive number of landslides (associated to floods and debrisflow). I don't know to quantify a exactly number of deaths associated directly to landslides but I am sure that this number was very elevated. Did you know about this event? If no, feel you free to ask me more about. Thank you, Francisco

---

## Referee Comment (RC2) · Anonymous Referee #2 · 17 Apr 2018

GENERAL COMMENT

The article "Global fatal landslide occurrence 2004 to 2016" presents a spatio-temporal analysis of a global dataset included in a continuously updated database. Several information are provided, also focusing on some hot spots and on some specific events. The description of the dataset is very detailed, while the description of the structure of the database could be improved. The theoretical background and the methodological approach are sufficiently explained. Figures are clear and explanatory, in particular the maps. The discussion regarding the connections among human actions and landslide fatalities is interesting.

The paper is quite long, especially in the sections 3.2 (Medium term trend in landslide occurrence) and 3.3 (Complex triggers). However, it is well written, in a good English

language. It is clear and easy to follow, except for the above mentioned sections that could be shortened. I suggest shortening the paper of about 10-15%.

The manuscript is interesting and the subject is within the topic of special issue of NHESS journal, definitely. If data had been made available to the public, it would have been an excellent strength of the work.

Overall, the manuscript is worth to be published. In my opinion, it needs minor/moderate revisions before being accepted.

In the following, I list some specific comments, concerning the terminology used in the paper and other issues. In addition, I have some comments on tables and figures, and I propose some other technical corrections.

SPECIFIC COMMENTS

First, I have several comments regarding the terminology used in the paper, which I believe should be ameliorated.

TERMINOLOGY:

In several places in the text, starting from the abstract (line 16), Authors write about "landslide events", without defining them. Given that in the literature several definitions of landslide events are proposed, I suggest introducing their definition at the beginning of the manuscript (e.g., in the introduction) and then use it consistently.

Moreover, in few parts of the manuscript, Authors use "databases" and "inventories" (e.g., line 40). Since they are not synonymous, I would suggest defining them and specifying the differences among them.

Further on in the text (line 51), Authors refer to "landslides and mass movements". Also in this case, I suggest specifying the differences, in any in the definition adopted.

At line 147, Authors define as "complex landslides" the landslides having a known trigger different from rainfall. Actually, "complex landslides" are one of the classes

Interactive
comment
defined by the most known and cited landslide classification (Varnes, 1978; Cruden and Varnes, 1996). For this reason, in order to avoid misunderstandings, I suggest choosing another definition.

In some places in the manuscript, precipitation and rainfall are used as synonymous. Actually, this is not exactly true, given that precipitation can include also snow or hailstorm. If the authors have precise information regarding rainfall and/or precipitation triggering of the landslides, an explanation would be interesting.

OTHER COMMENTS:

Line 27: regarding climate-landslides interactions, I suggest considering the recent work by Gariano and Guzzetti (2016), in which there are also several mentions to the feedbacks among climate, environmental, and human triggers (or disturbances, or impacts) on landslides that can be useful for the discussion at lines 442-452.

Line 60: please note that Dowling and Santi (2014) compiled a global catalogue of 213 debris flows that have resulted in 77,779 fatalities, in the period 1950–2011, recorded from academic publications, newspapers, and personal correspondence. The authors analysed spatial, temporal, and physical characteristics of the phenomena, and concluded stating that landslide mortality is higher in developing countries.

Moreover, the NatCat service and the Natural Hazards Assessment Network (NATHAN) prepared by Munich Re should be mentioned, even if they were designed for insurance purposes. Finally, I believe that some citations to national landslide catalogues and databases could be added in this section.

Lines 63-65: I suggest rewording the sentence at lines 63-65 to make it clearer.

Line 68: I suggest adding more information on the structure of the database, also from a technical point of view (e.g., used software(s), amount of data, data management).

Line 69: I suggest introducing an acronym for the "Global Fatal Landslide Database".

Lines 98-107: it seems to me that, in the database, some fields cataloguing the precision in the temporal definition and in the spatial localization of the landslides are missing. These would be very useful for data analysis, e.g., the analysis of the precision and the quality of the database, and also to support the statements at lines 106-107. An explanation is needed.

Line 111: I suggest adding a subsection title.

Line 140: I suggest rewriting the sentence to make it clearer. Sections 3.2 and 3.3 are very long. In some parts, they are very difficult to follow. I suggest a significant shortening of both subsections.

Line 428: In the final discussion, it should be acknowledged that such global database, as all global databases and catalogues, focuses primarily on major catastrophic events that typically have impacted large areas. For this reason, the number of the listed landslides is known to be largely underestimated, in particular regarding on low to medium intensity events, and on local events. The incompleteness of the global databases hampers their use for quantitative risk assessment studies, as reported in the mentioned article by Van Den Eeckhaut and Hervàs (2012).

Lines 442-447: I report in the following some statements extracted from the IPCC special report "Managing the Risks of Extreme Events and Disasters to Advance Climate Change Adaptation", in particular from Chapter 3 (Seneviratne et al., 2012), which could be useful for discussion:

- "Many weather and climate extremes are the result of natural climate variability (including phenomena such as El Niño), and natural decadal or multi-decadal variations in the climate provide the backdrop for anthropogenic climate changes. Even if there were no anthropogenic changes in climate, a wide variety of natural weather and climate extremes would still occur".

- "There is evidence that some extremes have changed as a result of anthropogenic

influences, including increases in atmospheric concentrations of greenhouse gases.".

- "There is low confidence in projections of an anthropogenic effect on phenomena such as shallow landslides in temperate and tropical regions, because these are strongly influenced by human activities such as poor land use practices, deforestation, and overgrazing".

TABLES

Table 1

The definition of landslide trigger for "precipitation" (Rainfall raises pore-pressure in slope materials triggering failure) is correct for deep-seated failures but could be slightly different for shallow phenomena as debris slides, debris avalanches, and debris flows. I suggest adding an explanation, following e.g., Sidle and Ochiai (2006).

Regarding "garbage collapse", I am somewhat doubtful about including this class in the analysis, given that these collapses do not pertain to natural slopes and consequently, given the different materials, the failure mechanisms are different. An explanation would be useful to the understanding.

FIGURES:

Figure 1: Just a comment, I find Figure 1very interesting and explicative. The comparison among the distribution of fatal landslides per Country and the location of most populated and richest areas in the globe is remarkable.

Figure 2: I would suggest adding the label "Year" for x-axis as done for Fig. 3. I would suggest adding tick marks also on left y-axis and on x-axis in correspondence to odd years (perhaps external to the graph).

Figure 3. I would suggest adding tick marks on x-axis in correspondence to odd years (maybe external to the graphs).

Figure 5. I would suggest using "J F M A M ..." instead of numbers for identifying

months. For a rapid understanding of the figure, labels "Mean daily rainfall (mm)" could be coloured in the same colours of the bars. Finally, why not using the same scale for all y-axes?

Figure 6. I would suggest adding tick marks on x-axis also in correspondence to odd years, to all panels.

Figure 7. The percentage for mining is missing. Moreover, in the please add total percentage instead of $\leq 1$ to collect all other classes.

Figure 8. I would suggest adding tick marks on x-axis also in correspondence to odd years, to all panels.

Figure 9. I would add "construction", "mining", "illegal mining", and "hill-cutting" in the four panels, above the legend, for a rapid understanding of the figure.

TECHNICAL CORRECTIONS

Line 19: "ENSO" should be defined for readers not expert in climate, meteorology and atmospheric sciences. Otherwise, I suggest deleting it from the abstract.

Line 29: please add "that" after "demonstrates".

Lines 62-62: please change "Kirschbaum et al., 2012; Kirschbaum et al., 2015" into "Kirschbaum et al., 2012; 2015".

Line 138: Hallegatte et al., 2018 is cited ad 2016 in the list.

Line 196: Magaña et al., 1999 is reported as 2003 in the reference list.

Line 238: Allen et al., 2015 is not in the reference list. In the list the paper is reported with the year 2016.

Line 240: replace "Fig. 7c" with "Fig. 6c". There are other wrong citation to Fig. 7 that should be corrected in Fig. 6 at lines 242, 244, 247, and 251.

Line 301: replace "Chan 1985; Chan 2000" with "Chan, 1985; 2000".

Line 324: Trenberth and Shea, 2005 is cited as "2006" in the reference list.

Line 371: replace "Yu and Li, 2011" with "Yu et al., 2011".

Line 421: Kumar and Pushplata, 2014 is cited as 2015 in the reference list.

Line 470: Kumar et al., 2018 is not in the reference list.

Line 482: please specify (a) or (b) at (World Bank, 2018).

Line 485: include "of" before "73".

Line 498: "Xie et al., 2013" is not in the reference list. There is a "Xie et al., 2003".

Line 499: "GPCC, 2018" is not in the reference list.

Lines 753-756: I have not found this reference cited in the text

REFERENCES

Cruden D.M., Varnes, D.J.: Landslide types and processes, in: Turner, A.K., Schuster, R.L. (Eds.), Landslides, Investigation and Mitigation, Special Report 247. Transportation Research Board, Washington D.C., pp. 36–75 ISSN: 0360-859X, ISBN: 030906208X, 1996.

Dowling C.A., Santi P.M.: Debris flows and their toll on human life: a global analysis of debris-flow fatalities from 1950 to 2011, Natural Hazards, 71, 1, 203-227, doi:10.1007/s11069-013-0907-4, 2014.

Gariano S.L., Guzzetti F.: Landslides in a changing climate. Earth-Science Reviews, 162, 227–252, doi: 10.1016/j.earscirev.2016.08.011, 2016.

Seneviratne S.I., Nicholls N., Easterling D., Goodess C.M., Kanae S., Kossin J., Luo Y., Marengo J., McInnes K., Rahimi M., Reichstein M., Sorteberg A., Vera C., Zhang X.: Changes in climate extremes and their impacts on the natural physical environment, in: Field, C.B., Barros, V., Stocker, T.F., Qin, D., Dokken, D.J., Ebi, K.L., Mastrandrea, M.D., Mach, K.J., Plattner, G.-K., Allen, S.K., Tignor, M., Midgley, P.M. (Eds.) Managing

the Risks of Extreme Events and Disasters to Advance Climate Change Adaptation. A Special Report of Working Groups I and II of the Intergovernmental Panel on Climate Change (IPCC). Cambridge University Press, Cambridge, UK, and New York, NY, USA, pp. 109–230, 2012.

Sidle R.C., Ochiai H.: Landslides: processes, prediction, and land use. Water Resour. Monogr. Ser. 18. AGU, Washington DC. 312 pp., doi:10.1029/WM018, 2006.

Van Den Eeckhaut M., Hervás J.: State of the art of national landslide databases in Europe and their potential for assessing landslide susceptibility, hazard and risk, Geomorphology , 139-140, 545–558, doi:10.1016/j.geomorph.2011.12.006, 2012.

Varnes D.J.: Slope movement types and processes., im: Schuster RL, Krizek RJ (Eds.) Landslides, analysis and control, special report 176: Transportation research board, National Academy of Sciences, Washington, DC., pp. 11–33, 1978.

---

## Author Comment (AC1) · 14 May 2018

*We thank the reviewer for their comments and constructive suggestions in their review of manuscript number nhess-2018-49. Please find below the authors' replies (in italics) to these comments:*

The article "Global fatal landslide occurrence 2004 to 2016" presents a spatio-temporal analysis of a global dataset included in a continuously updated database. Several information are provided, also focusing on some hot spots and on some specific events. The description of the dataset is very detailed, while the description of the structure of the database could be improved. The theoretical background and the methodological approach are sufficiently explained. Figures are clear and explanatory, in particular the

maps. The discussion regarding the connections among human actions and landslide fatalities is interesting. The paper is quite long, especially in the sections 3.2 (Medium term trend in landslide occurrence) and 3.3 (Complex triggers). However, it is well written, in a good English language. It is clear and easy to follow, except for the above mentioned sections that could be shortened. I suggest shortening the paper of about 10-15

*\*\*The authors are in the process of developing a web platform to host data*

SPECIFIC COMMENTS

First, I have several comments regarding the terminology used in the paper, which I believe should be ameliorated.

TERMINOLOGY:

In several places in the text, starting from the abstract (line 16), Authors write about "landslide events", without defining them. Given that in the literature several definitions of landslide events are proposed, I suggest introducing their definition at the beginning of the manuscript (e.g., in the introduction) and then use it consistently. *A landslide event is a single slope failure. Landslide reports typically differentiated the number of individual fatal landslides in a multiple occurrence regional landslide event (MORLE, Crozier, 2005).*

Moreover, in few parts of the manuscript, Authors use "databases" and "inventories" (e.g., line 40). Since they are not synonymous, I would suggest defining them and specifying the differences among them. *The term 'database' is most appropriate. An inventory is a 'complete' list, while a database is a structured set of data. The term inventory will be replaced in text with database.*

Further on in the text (line 51), Authors refer to "landslides and mass movements". Also in this case, I suggest specifying the differences, in any in the definition adopted. *Remove "landslides and"*

[Figure]

At line 147, Authors define as "complex landslides" the landslides having a known trigger different from rainfall. Actually, "complex landslides" are one of the classes defined by the most known and cited landslide classification (Varnes, 1978; Cruden and Varnes, 1996). For this reason, in order to avoid misunderstandings, I suggest choosing another definition. *The authors agree this term could cause confusion. The authors suggest the use of an acronym for non-seismic non-rainfall triggered landslides (NSNR).*

In some places in the manuscript, precipitation and rainfall are used as synonymous. Actually, this is not exactly true, given that precipitation can include also snow or hailstorm. If the authors have precise information regarding rainfall and/or precipitation triggering of the landslides, an explanation would be interesting. *The authors will replace "precipitation" with "rainfall"*

OTHER COMMENTS:

Line 27: regarding climate-landslides interactions, I suggest considering the recent work by Gariano and Guzzetti (2016), in which there are also several mentions to the feedbacks among climate, environmental, and human triggers (or disturbances, or impacts) on landslides that can be useful for the discussion at lines 442-452. *The work by Gariano and Guzzeti (2016) is indeed a comprehensive and an interesting review of research on the impacts of climate change on landslides. Suggestions for revision to paragraph starting line 442 are included in the response to anonymous referee 1. The authors will add the sentence "Gariano and Guzzetti (2016) provide a comprehensive review of research on the impacts of climate change on landslides; particularly highlighting the effect of different climate variables on different landslide types." before "fatal landslides" on line 447.*

Line 60: please note that Dowling and Santi (2014) compiled a global catalogue of 213 debris flows that have resulted in 77,779 fatalities, in the period 1950–2011, recorded from academic publications, newspapers, and personal correspondence. The authors

analysed spatial, temporal, and physical characteristics of the phenomena, and concluded stating that landslide mortality is higher in developing countries. *Dowling and Santi's (2014) study will be acknowledged after the sentence ending on line 63.*

Moreover, the NatCat service and the Natural Hazards Assessment Network (NATHAN) prepared by Munich Re should be mentioned, even if they were designed for insurance purposes. Finally, I believe that some citations to national landslide catalogues and databases could be added in this section. *Other databases will be acknowledged on line 63, including NatCat.*

Lines 63-65: I suggest rewording the sentence at lines 63-65 to make it clearer *The term 'complex' landslides will be replaced with an acronym for non-seismic non-rainfall (NSNR) triggered landslides. This should remove confusion with Varnes' classification of landslide type.*

Line 68: I suggest adding more information on the structure of the database, also from a technical point of view (e.g., used software(s), amount of data, data management). *A sample of the database may be included in the appendix material to provide insight into the structure. With this sample, the authors will provide metrics on the number of entries, file size, software used to manage the data. A web platform is currently under development to host the database open source.*

Line 69: I suggest introducing an acronym for the "Global Fatal Landslide Database". *An acronym can be used. Simply GFLD may be most appropriate.*

Lines 98-107: it seems to me that, in the database, some fields cataloguing the precision in the temporal definition and in the spatial localization of the landslides are missing. These would be very useful for data analysis, e.g., the analysis of the precision and the quality of the database, and also to support the statements at lines 106-107. An explanation is needed *This paper uses national political boundaries as the lowest spatial unit for analysis. All landslide entries can be located within a national boundary. The spatial precision of each landslide report varies depending on the quality of*

*reporting. The description of the location of the landslide may provide the name of the village in which a landslide occurred, or the stretch of road affected. Some reports are less specific and simply include the broader administrative zone (e.g. state, county).*

*When the landslide is reported in an administrative zone (village, city, stage), the administrative boundaries are used. For example, if the landslide occurred in the city of Kathmandu, then the administrative boundaries for Kathmandu define the spatial precision for the landslide report. These are extracted from the GADM database (2018). If the landslide occurred on a stretch of road network, then the stretch of road is mapped in GIS and a buffer applied to include slopes along the road. Some landslides can be identified in Google Earth imagery or Planet Labs imagery and these are individually mapped in GIS, so that the precise location of the landslide is known.*

*1-2 sentences will be added to the manuscript to provide more detail on the spatial precision.*

Line 111: I suggest adding a subsection title. *The authors could add a sub-title "3.1. Global overview"*

Line 140: I suggest rewriting the sentence to make it clearer. *The authors do not believe this sentence requires rewording.*

Sections 3.2 and 3.3 are very long. In some parts, they are very difficult to follow. I suggest a significant shortening of both subsections. *The authors disagree with this recommendation. The authors recognise the manuscript is long, however the sections have been written to not only present results from the global fatal landslide database, but also promote discussion and further research within the landslide community. In Section 3.2, the medium-term trends in the data (trends over several years) are discussed. The peak in global landslide activity between 2009 and 2011, documented by Kirschbaum et al. (2012; 2015), is associated with a moderate El Nino and La Nina. Within Section 3.2 we summarise the regional patterns of rainfall responsible for increased landslide activity between 2009 and 2011, and how rainfall is related to*

*ENSO phase. Considering other moderate to strong ENSO events, the database does not find a consistent relationship between ENSO phase and regional landslide occurrence. This analysis is by no means definitive, rather it aims to highlight that further research is required to link regional landslide occurrence with global climate. Section 3.3 aims to contextualise the observed increase in construction and mining triggered landslides, by associating the increase with observations from recent research on landslide impact. The section not only presents the overall trend in data, but uses the depth of information provided by landslide reports to identify recurrent circumstances in which fatal landslides occur and what may be driving specific increases. Often there is very little existing research on these local contexts, and it is hoped that this paper inspires future work.*

Line 428: In the final discussion, it should be acknowledged that such global database, as all global databases and catalogues, focuses primarily on major catastrophic events that typically have impacted large areas. For this reason, the number of the listed landslides is known to be largely underestimated, in particular regarding on low to medium intensity events, and on local events. The incompleteness of the global databases hampers their use for quantitative risk assessment studies, as reported in the mentioned article by Van Den Eeckhaut and Hervàs (2012). *The limitations of the database have been acknowledged in within section 2 (specifically lines 81 to 97). Petley (2012) estimated the true level of loss to be underestimated by 15 per cent. The limits of the database will be emphasised again in the discussion section. Critically, global studies using hazard databases collated from media reporting are designed to capture general trends in landslide occurrence rather than provide data for local quantitative risk assessment. Results presented from this database should provide a catalyst for future research aimed at understanding the complete picture of local landslide impact. With the growing permeation of telecommunication technology to remote mountainous regions, it is expected that landslide reporting will increase, improving the completeness of the global database.*

Lines 442-447: I report in the following some statements extracted from the IPCC special report "Managing the Risks of Extreme Events and Disasters to Advance Climate Change Adaptation", in particular from Chapter 3 (Seneviratne et al., 2012), which could be useful for discussion: "Many weather and climate extremes are the result of natural climate variability (including phenomena such as El Niño), and natural decadal or multi-decadal variations in the climate provide the backdrop for anthropogenic climate changes. Even if there were no anthropogenic changes in climate, a wide variety of natural weather and climate extremes would still occur". "There is evidence that some extremes have changed as a result of anthropogenic influences, including increases in atmospheric concentrations of greenhouse gases.". "There is low confidence in projections of an anthropogenic effect on phenomena such as shallow landslides in temperate and tropical regions, because these are strongly influenced by human activities such as poor land use practices, deforestation, and overgrazing". *A general statement on the challenges of differentiating the impact of climate and anthropogenic effects on global landslide occurrence (with reference to the IPCC report) will be included on line 441 of discussion.*

TABLES

Table 1 The definition of landslide trigger for "precipitation" (Rainfall raises pore-pressure in slope materials triggering failure) is correct for deep-seated failures but could be slightly different for shallow phenomena as debris slides, debris avalanches, and debris flows. I suggest adding an explanation, following e.g., Sidle and Ochiai (2006). *This will be clarified in the table.*

Regarding "garbage collapse", I am somewhat doubtful about including this class in the analysis, given that these collapses do not pertain to natural slopes and consequently, given the different materials, the failure mechanisms are different. An explanation would be useful to the understanding *The database contains a number of failures in garbage piles, frequently in large garbage accumulations (city rubbish dumps). Although these failures occur in non-natural materials, similar mitigation measures (as on*

natural slopes) need to be considered. These include appropriate stacking of material to account for stable slope design, barriers to protect workers and restricted access to sites during heavy rainfall. Furthermore, slope failures in municipal solid waste dumps and landfills have been discussed by engineers using slope stability models designed for slopes composed of natural materials. For these reasons, we include garbage collapse landslides in the database.

FIGURES:

Figure 1: Just a comment, I find Figure 1 very interesting and explicative. The comparison among the distribution of fatal landslides per Country and the location of most populated and richest areas in the globe is remarkable.

Figure 2: I would suggest adding the label "Year" for x-axis as done for Fig. 3. I would suggest adding tick marks also on left y-axis and on x-axis in correspondence to odd years (perhaps external to the graph). *The x-axis title "Year" will be added. Tick marks will be added for the right-hand axis (Cumulative number of landslides). Tick marks will be moved to 'outside' the graph on the x-axis.*

Figure 3. I would suggest adding tick marks on x-axis in correspondence to odd years (maybe external to the graphs). *Tick marks will be moved to 'outside' the graph on the x-axis.*

Figure 5. I would suggest using "J F M A M : : :" instead of numbers for identifying months. For a rapid understanding of the figure, labels "Mean daily rainfall (mm)" could be coloured in the same colours of the bars. Finally, why not using the same scale for all y-axes? *The labelling will be changed as suggested. The axis scale will not change. The mean daily landslide occurrence varies by an order of magnitude between regions.*

Figure 6. I would suggest adding tick marks on x-axis also in correspondence to odd years, to all panels. *Tick marks will be moved to 'outside' the graph on the x-axis.*

Figure 7. The percentage for mining is missing. Moreover, in the please add total

percentage instead of 1 to collect all other classes. *The percentage for mining (unknown) will be added and the total percentage of the human activity…etc classes will be included instead of <1.*

Figure 8. I would suggest adding tick marks on x-axis also in correspondence to odd years, to all panels. *Tick marks will be moved to 'outside' the graph on the x-axis.*

Figure 9. I would add "construction", "mining", "illegal mining", and "hill-cutting" in the four panels, above the legend, for a rapid understanding of the figure. *Labels will be added to each map.*

TECHNICAL CORRECTIONS

Line 19: "ENSO" should be defined for readers not expert in climate, meteorology and atmospheric sciences. Otherwise, I suggest deleting it from the abstract. *ENSO will be defined in the abstract (line 19) and text (line 282)*

Line 20: please add "that" after "demonstrates". *"That" will be added*

Lines 62-62: please change "Kirschbaum et al., 2012; Kirschbaum et al., 2015" into "Kirschbaum et al., 2012; 2015". *This will be modified.*

Line 138: Hallegatte et al., 2018 is cited ad 2016 in the list. *This will be amended to 2016 on line 138*

Line 196: Magaña et al., 1999 is reported as 2003 in the reference list. *The 2003 reference will be removed and replaced with Magaña, V., Amador, J. A. and Medina, S.: The Midsummer Drought over Mexico and Central America, J. Clim., 12(6), 1577–1588, doi:10.1175/1520-0442(1999)012<1577:TMDOMA>2.0.CO;2, 1999.*

Line 238: Allen et al., 2015 is not in the reference list. In the list the paper is reported with the year 2016. *This will be amended to 2016 on line 238*

Line 240: replace "Fig. 7c" with "Fig. 6c". There are other wrong citation to Fig. 7 that should be corrected in Fig. 6 at lines 242, 244, 247, and 251. *Typographical error: Line*

*240- "Fig 7c" to Fig 6c. Line 242- "Fig 7d" to Fig 6d. Line 244- "Fig 7c and 7d" to Fig 6c and 6d. Line 244- "Fig. 7e" to Fig 6e. Line 247- "Fig. 7f" to Fig 6f. Line 247- "Fig. 7g" to Fig 6g. Line 251- "Fig. 7f and 7g" to Fig 6f and 6g.*

Line 301: replace "Chan 1985; Chan 2000" with "Chan, 1985; 2000". *This edit will be made.*

*New references*

*Crozier, 2005. Multiple-occurrence regional landslide events in New Zealand: hazard management issues. Landslides 2(4), p.247-256*

---

## Author Comment (AC2) · 14 May 2018

*We thank the reviewer for their comments and constructive suggestions in their review of manuscript number nhess-2018-49. Please find below the authors' replies (in italics) to these comments:*

Summary: Overall this manuscript is well written, provides interesting findings reviewing over a decade of landslide reports and adds value to this topic within the landslide community. With minor revisions, I recommend this manuscript for publication. I do not see any major weaknesses in the methodology or evaluation methods and appreciate the insight provided in the discussion section regarding the challenge with anthropogenic modification of hillslopes in causing landslide fatalities; however the authors

state in several places that human disturbance may be more detrimental to future landslide incidences than climate. I think a more accurate statement is that it may be more detrimental to future fatal landslides incidences, as the understanding of how climate may modulate change in extreme rainfall and how that may impact landsliding has not, to my knowledge, been robustly addressed. Please see specific comments below.

Line 106-7: How was the spatial precision determined? While this does not affect the analysis in the paper, it would be helpful to provide another sentence or a reference to how this precision was obtained for each report.

*The description of the location of the landslide is extracted from each media report. Many entries provide the name of the village in which a landslide occurred, or the stretch of road affected. Some reports are less specific and simply include the broader administrative zone (e.g. state, county). All landslides are known to country level.*

*The spatial precision of each landslide report varies depending on the quality of reporting. When the landslide is reported in an administrative zone (village, city, stage), the administrative boundaries are used. For example, if the landslide occurred in the city of Kathmandu, then the administrative boundaries for Kathmandu define the spatial precision for the landslide report. These are extracted from the GADM database (2018). If the landslide occurred on a stretch of road network, then the stretch of road is mapped in GIS and a buffer applied to include slopes along the road. Some landslides can be identified in Google Earth imagery or Planet Labs imagery and these are individually mapped in GIS, so that the precise location of the landslide is known.*

*1-2 sentences will be added to the manuscript to provide more detail on the spatial precision.*

Line 171: What precipitation data did you use in your analysis? *The rainfall data was from the Global Precipitation Climatology Centre data (Xie et al., 2013; GPCC, 2018). This is stated in the caption for Figure 5.*

Line 195: I believe there is a word missing from the sentence that starts with "The season is bimodal: : :" possibly "with peaks in rainfall on either side of a midsummer: : :" *This sentence will be amended as follows.*

*Original sentence: "The season is bimodal, with peaks in rainfall either side of a midsummer drought, between late June to August (Magaña et al., 1999)."*

*Amended sentence: "The season is bimodal, with peaks in rainfall on either side of a midsummer drought, between late June to August (Magaña et al., 1999)."*

Lines 240, 244, 247, 251: Should be changed to Figure 6, not Figure 7 *Typographical error: Line 240- "Fig 7c" to Fig 6c. Line 242- "Fig 7d" to Fig 6d. Line 244- "Fig 7c and 7d" to Fig 6c and 6d. Line 244- "Fig. 7e" to Fig 6e. Line 247- "Fig. 7f" to Fig 6f. Line 247- "Fig. 7g" to Fig 6g. Line 251- "Fig. 7f and 7g" to Fig 6f and 6g.*

Line 261: What is meant by the term unsettled weather?

*I cite NOAA's (2018) definition here for "unsettled": "In meteorological use: A colloquial term used to describe a condition in the atmosphere conducive to precipitation. This term typically is associated with the passage of surface or upper level low pressure systems, fronts or other phenomenon when precipitation expected."*

*Unsettled weather is associated with unstable air masses. Unstable atmospheric conditions occur when air in the lower levels of the atmosphere is warm (and/or humid) and rises, and continues to rise by being warmer than its surrounding environment. The rise of hot air leaves lower pressure regions, into which air from high pressure moves, heats and then rises. Conditions favouring this convection occur when sea surface temperatures are warmer. Not all low pressure systems may be large enough to be classified on the hurricane (or typhoon) scale, but these smaller systems, may still be associated with significant rainfall.*

*In the database, many landslides were triggered in hurricane prone regions, during hurricanes, by smaller low pressure systems.*

*To improve clarity the authors amend "likely due to unsettled weather associated with the passage of large storms in the region." with… likely due to unsettled weather associated with warm sea surface temperatures in the region.*

Line 312: Suggest changing "which" to "that" and removing the comma

*Agreed, this edit will be made.*

Lines 439-441: Run on sentence, consider revising.

*This sentence will be amended as follows:*

*Original sentence: "Future work bridging advances in climate science on regional impacts from ENSO diversity, with local patterns of landsliding, in acutely affected areas such as India, China and Nepal, will provide useful models for forecasting seasonal rainfall distribution and landslide impact."*

*Amended sentence: "Increased understanding of the impact of ENSO diversity on regional climate, will improve models forecasting seasonal rainfall distribution and landslide impact. This is particularly important in acutely affected areas such as India, China and Nepal."*

Line 444: Consider adding "fatal" to the wording, "disturbance (land use change) may be more detrimental to future fatal landslides incidence: : :"

*In this particular sentence, it is more appropriate to remain broad, as the studies referenced (line 444) consider landslide occurrence (number) rather than landslide impacts.*

*This paper does not aim to evaluate whether human disturbances or climate change may be more detrimental to landslide incidence. That question requires a longer time series of data ( 30 years) and we highlight this in the paper. The results of this study show that fatal landslides triggered by human activities are increasing (2004 to 2016). Specifically due to mining and construction.*

*To improve clarity we will amend the following: Swap sentence line 442 "our analyses. . . hillcutting" and line 443 "human disturbance. . .2015)". Add "fatal" before "landslide occurrence" line 442. Modify "in particular in relation to" with "driven by" line 443.*

Line 453- 55: Suggest reworking sentence, it reads rather awkwardly

*Original sentence: "Holcombe et al. (2016) emphasised that planning policy alone is not sufficient to control landslide risk in developing nations, because of the rapid and informal nature of construction, and low-income of residents, who cannot finance expert guidance when building their homes."*

*Amended sentence: "Holcombe et al. (2016) emphasised that planning policy alone is not sufficient to control landslide risk in developing nations. This is due to the rapid and informal nature of construction, and low-income of residents, who cannot finance expert guidance when building their homes."*

Line 460-461: Sentence starting with "Several landslides were triggered: : :" is a repetitive sentence and is likely not needed here.

*The sentence highlights a specific circumstance by which people are killed by hillcutting triggered landslides. This sentence should remain.*

Line 461: Suggest omitting the word "simple" as often effective communication of risk by the government is not that simple.

*Agreed, omit simple*

Lines 471-476: Suggest reworking sentence, it reads rather awkwardly

*Original sentence: "The dataset is a useful tool in identifying acutely landslide prone parts of the world and specific local drivers of landslide impact; thereby highlighting locations which would benefit from further development in early warning technology, landslide risk assessment and community capacity building; in support of the future directions of the International Consortium on Landslides (Alcantara- Ayala et al., 2017)."*

*Amended sentence: "The dataset is a useful tool in identifying acutely landslide prone parts of the world and specific local drivers of landslide impact; thereby highlighting locations which would benefit from further development in early warning technology, landslide risk assessment and community capacity building. This is in support of the future directions of the International Consortium on Landslides (Alcantara- Ayala et al., 2017)."*

Line 479 and other references: is it necessary to write AD prior to the years 2004 to 2016? I suggest removing as I am confident readers will not confuse it with events 4000 years ago.

*The reference to AD will be removed*

*References*

*GADM (2018) https://gadm.org/index.html, accessed 25/4/18*

*NOAA (2018) https://forecast.weather.gov/glossary.php?word=unsettled, accessed 25/4/18*

*Xie, P., Janowiak, J. E., Arkin, P. A., Adler, R., Gruber, A., Ferraro, R., Huffman, G. J. and Curtis, S.: GPCP Pentad Precipitation Analyses: An Experimental Dataset Based on Gauge Observations and Satellite Estimates, J. Clim., 16(13), 2197–2214, doi:10.1175/2769.1, 2003.*

*GPCC (2018) https://www.esrl.noaa.gov/psd/data/gridded/data.gpcc.html, accessed 25/4/18*

---

## Author Comment (AC3) · 14 May 2018

The authors thank Jan Klimes for his comments on the manuscript.

The authors agree that it is highly important for the observations and predictions of landslide scientists to be acted on by society. The article serves to highlight several contexts in which landslide mitigation can reduce harm to people.

The authors recognise that the proportion of fatal landslides that go unreported varies by country. Sepulveda and Petley (2015) demonstrated that 95% of fatal landslide reports in South and Central America were captured by the database using an English based language search, compared with using a Spanish and Portuguese based language search. It would be useful to repeat this exercise in the future for Asia.

A valid point is made on line 454 that planning policies alone are not sufficient to control landslide risk in developed nations, as well as developing nations. The Litochovice landslide (Czech Republic, 2013) mentioned in the comment may have been preconditioned by road construction at the landslide toe and mining at the head of the landslide. Prior geomorphological mapping and slope assessment to identify potential slope instability is an important step in infrastructure development that can be guided by policy.
* * *

---

## Author Comment (AC4) · 14 May 2018

Thank you for your comment.

In total we registered 1748 fatalities from 16 landslides in Brazil in 2011. Of these 6 landslides (killing 817 people) occurred in Rio de Janeiro State. Two multiple occurrence regional landslide events (MORLE, Crozier, 2005) occurred in 2011 killing a total of 836 people.

The comment highlights an important point that these MORLEs should be differentiated in the analysis. The results will be modified. MORLEs account for less than 5% of entries in the database and do not effect overall trends in the data. Landslides reported as a single MORLE result in an underestimate of the true number of individual landslide

events.

Reference Crozier, 2005. Multiple-occurrence regional landslide events in New Zealand: hazard management issues. Landslides 2(4), p.247-256
* * *